# Multivariate Multiscale Cosine Similarity Entropy and Its Application to Examine Circularity Properties in Division Algebras [note 1]

**DOI:** 10.3390/e24091287

**Published:** 2022-09-13

**Authors:** Hongjian Xiao, Theerasak Chanwimalueang, Danilo P. Mandic

**Affiliations:** 1Department of Electrical and Electronic Engineering, Imperial College London, London SW7 2AZ, UK; 2Faculty of Engineering, Srinakharinwirot University, Nakhon Nayok 26120, Thailand

**Keywords:** cosine similarity entropy, angular distance, multi-channel system, detection of circularity, complex circularity, quaternion circularity, multivariate entropy

## Abstract

The extension of sample entropy methodologies to multivariate signals has received considerable attention, with traditional univariate entropy methods, such as sample entropy (SampEn) and fuzzy entropy (FuzzyEn), introduced to measure the complexity of chaotic systems in terms of irregularity and randomness. The corresponding multivariate methods, multivariate multiscale sample entropy (MMSE) and multivariate multiscale fuzzy entropy (MMFE), were developed to explore the structural richness within signals at high scales. However, the requirement of high scale limits the selection of embedding dimension and thus, the performance is unavoidably restricted by the trade-off between the data size and the required high scale. More importantly, the scale of interest in different situations is varying, yet little is known about the optimal setting of the scale range in MMSE and MMFE. To this end, we extend the univariate cosine similarity entropy (CSE) method to the multivariate case, and show that the resulting multivariate multiscale cosine similarity entropy (MMCSE) is capable of quantifying structural complexity through the degree of self-correlation within signals. The proposed approach relaxes the prohibitive constraints between the embedding dimension and data length, and aims to quantify the structural complexity based on the degree of self-correlation at low scales. The proposed MMCSE is applied to the examination of the complex and quaternion circularity properties of signals with varying correlation behaviors, and simulations show the MMCSE outperforming the standard methods, MMSE and MMFE.

## 1. Introduction

The investigation of entropy has been a considerably important topic in nonlinear analysis, especially in relation to the quantification of the degree of irregularity and multiple correlations within time series. Entropy-based complexity studies have been implemented in many physical and physiological studies, mostly based on the widely accepted complexity loss theory (CLT) [1]. The CLT states that the physiological responses of the human body in a pathological state (resulting from, for example, stress, illness, or aging) exhibit a loss in structural complexity, while the highest degree of irregularity is associated with healthy and young bio-systems [1,2].

The concept of entropy derives from thermodynamics to measure the probability of micro-states in physics [3]. When it comes to Shannon’s information theory, entropy is employed to describe the amount of information and the generation rate of new information [4,5]. Among numerous methods to characterize signals in terms of entropy, two commonly used techniques are approximate entropy [6] and sample entropy [7]. Both employ the Chebyshev distance and utilize estimates of the conditional probability of similar patterns within the time series. Sample entropy is usually considered an enhancement of approximate entropy, as it gives less biased estimation and higher robustness [7]. Subsequently, sample entropy (SampEn) has found applications in both physical and physiological systems [8,9,10,11,12,13]. For example, research based on SampEn of the electrocardiogram (ECG) has been utilized to estimate different degrees of atrial fibrillation [14], or the analysis of stress in public speaking [15]. A further improvement of SampEn, called fuzzy entropy (FuzzyEn), employs fuzzy functions as a substitute for the hard thresholding within SampEn in the definition of similar patterns [16]. Given a continuous relation between the similarity and the obtained distance, FuzzyEn is capable of giving a more reliable and well-defined estimation, while requiring shorter data length compared to SampEn [16]. Moreover, FuzzyEn has been applied in research on the dynamics of gaze complexity during virtual sailing [17].

Despite the broad use of SampEn and FuzzyEn, their shortcomings still remain when it comes to real-world data. The problems include the following:The distance measure utilized in SampEn and FuzzyEn is based on the amplitude of the time series, where the unlimited range would lead to undefined values.Due to the signal related parameter setting, when the tolerance is set according to the value of standard deviation, the data normalization is required in advance, which would re-scale the original signal and cause information loss.Considering the requirement of re-scaling, this affects the noise robustness of SampEn and FuzzyEn, in particular, when processing short time series.Long data length is required for traditional SampEn and FuzzyEn analyses when exploring complexity with a high embedding dimension.

Recently, a new theory associated with self-correlation states that pathology is simultaneously followed with an increase in structural complexity (e.g., correlation within time series), which can be considered a complementary theory to the traditional complexity loss theory (CLT) [18]. This promises to help resolve the problems with SampEn- and FuzzyEn-based methodologies, which fail to give a holistic quantification of the self-correlation of the target signals. Unlike other existing entropy methods, cosine similarity entropy (CSE) was introduced to quantify the structural complexity of time series in terms of self-correlations based on angular distance [18].

The CSE provides a robust and more meaningful estimation of the structural complexity of chaotic systems, in contrast to SampEn and FuzzyEn [18]. The extension of CSE to multivariate case is introduced in this paper, whereby the proposed multivariate multiscale cosine similarity entropy (MMCSE) is based on the idea of the composite delay vector introduced in multivariate multiscale sample entropy (MMSE) [19] and the rule of angular distance applied in CSE [18]. The performance of MMCSE is first evaluated over all parameter settings, and then applied to the detection of complex and quaternion non-circularity in multi-channel systems, whereby different correlation behaviors were examined and discussed by multivariate entropy methods. The virtues of MMCSE are further illustrated by correctly quantifying the meaningful structural complexity for varying conditions of correlation degrees and power in data channels.

The reminder of the paper is organized as follows. Related work and development of SampEn and FuzzyEn-based entropy are reviewed in Section 2. To contrast the structure of the cosine similarity entropy method, traditional sample entropy and fuzzy entropy along with their multivariate extended forms are presented and reviewed in Section 3. The details of the extension of cosine similarity entropy to the multivariate case are given in Section 4. Section 5 discusses the selection of parameters and provides the suggested values for the manually set parameters when applying multivariate multiscale cosine similarity entropy. Section 6 illustrates the application of MMCSE in the detection of circularity in terms of varying correlation behaviors. Finally, the conclusions are given in the last section.

## 2. Related Work and Development of Existing Sample Entropy and Fuzzy Entropy

Although FuzzyEn exhibits a higher consistency at the expense of computational load, single-scale processing has shown limitations in real-world entropy analysis. Hence, the ’coarse-graining process’ (CGP) was subsequently introduced in traditional entropy methodologies to yield multiscale entropy [20], which allows for the examination of information at high scales. This is achieved by taking the average over neighboring data points. However, it was realized that CGP is not the optimal method to obtain the scaled signal because the traditional CGP in the frequency domain operates as a low-pass filter with large stop band ripples, resulting in biased scaled signals containing artifacts [21]. In addition, the averaging procedure of CGP requires a large data set and comes with a loss of the high-frequency component in data [22]. To overcome the first drawback of CGP, refined multiscale entropy was introduced by employing a low-pass Butterworth filter to generate unbiased scaled signals [21]. As a refinement of CGP in terms of the second flaw, several improved methods have been proposed. For instance, composite multiscale sample entropy operates by implementing CGP at various starting points and taking the average over the resulting entropy values [23], while generalized multiscale entropy employs higher moments (e.g., variance) to give the dynamics over multiple scales in place of the mean [24]. Another multiscale strategy that has been applied in entropy analysis is named intrinsic mode entropy (IMEn), and employs intrinsic mode functions (IMFs) obtained by empirical mode decomposition (EMD), whereby the scaled signals in IMEn are obtained by cumulative sums of IMFs to introduce robustness toward low-frequency changes [25]. However, the additional processes used to improve CGP inevitably increase the computational time. Despite these drawbacks of CGP, it is widely used in real-world applications since it balances computation efficiency with reasonable entropy estimates.

Multivariate multiscale sample entropy (MMSE) was proposed as an extension to SampEn in order to simultaneously process information from multiple channels and relax the requirement of long data length for uni-variate entropy [19]. For multi-channel data, MMSE provides a more powerful and accurate estimation, even for limited data length [19]. Multivariate multiscale fuzzy entropy (MMFE) was subsequently introduced, which maintains the merits of FuzzyEn in contrast to SampEn [26], in the multivariate setting. Both FuzzyEn- and SampEn-based methods are built based on the concept of amplitude distance, whereby outliers manifest themselves as spurious peaks and have a severe impact on the accuracy of estimation of structural complexity [18]. Further, the complexity quantified by SampEn or FuzzyEn is based on randomness and irregularity, which is one aspect of complex systems, while other properties, such as long-range correlations, fail to be demonstrated by single-scale SampEn and FuzzyEn [27]. Note that the scale of interest in multiscale entropy, based on amplitude distance, has no unified measure indicators as criteria when the scale is guaranteed to be set as a certain value to exhibit the long-range correlation of the signal.

## 3. Multivariate Multiscale Sample Entropy and Multivariate Multiscale Fuzzy Entropy

In this section, two traditional entropies, sample entropy (SampEn) and fuzzy entropy (FuzzyEn), are introduced and compared. Starting from a uni-variate single-scale process to the multivariate multi-scale approach, both SampEn and FuzzEn are developed based on the amplitude distance and the conditional probability of similar patterns.

### 3.1. Sample Entropy and Fuzzy Entropy

The details of sample entropy are given in Algorithm 1. In standard SampEn, the original signal is first reconstructed in the phase space following Takens’ embedding theory [28]. Each embedding vector is jointly described by the embedding dimension, *m*, and time delay, *l*. Then, the Chebyshev distance is applied to pairs of the embedding vectors to give the degree of similarity, where a similar pair is defined by the Heaviside function with the tolerance coefficient, *r*. In the final stage, the probability of similar patterns is produced, and the estimated result is generated by the ratio of the probabilities given in increased phase spaces.

Similarly to sample entropy, fuzzy entropy employs the Chebyshev distance to give the similarity between paired embedding vectors. Here, the embedding matrix needs to be centered prior to the reconstruction. Next, the embedding matrix, $X_{m}\left(i\right)$, is formed and the distance matrix $D_{m}\left(i,j\right)$ is computed. The details of FuzzyEn are presented in Algorithm 2. The main modification of FuzzyEn from SampEn is the involvement of a continuous transformation from the distance, $d_{m}\left(i,j\right)$, to the similarity, $s_{m}^{r_{f}}\left(i,j\right)$, through the fuzzy membership functions as given in the Step 3 in Algorithm 2. Benefiting from the employment of the fuzzy membership functions, undefined estimation due to the absence of matching pairs is largely avoided. However, a higher computational load is required compared to SampEn, and the performance of FuzzyEn generally shows improved but also similar trends as with SampEn.
**Algorithm 1.** Sample EntropyGiven a univariate data set ${\left\{x\left(i\right)\right\}}_{i=1}^{N}$ of length *N*,  the parameters involved are the embedding dimension, *m*, tolerance, $r_{s}$, and time delay, *l*.
Construct the embedding matrix, $X_{m}$, derived from the original signal, ${\left\{x\left(i\right)\right\}}_{i=1}^{N}$, in the form $x_{m}\left(i\right)=\left(x\left(i\right),x\left(i+l\right),\ldots ,x\left(i+\left(m-1\right)l\right)\right)$.Compute the distance between all the pairwise embedding vectors, $x_{m}\left(i\right)\;\;\&\;\;x_{m}\left(j\right)$, based on the Chebyshev distance, as $d_{m}\left(i,j\right)=max\{x_{m}\left(i+k\right)-x_{m}\left(j+k\right)||\;0\leq k\leq m-1,i\neq j\}$.Compute the number of matching patterns, $B_{m}^{r_{s}}\left(i\right)$, defined as the similar pairs that satisfy the criterion $d_{m}\left(i,j\right)\leq r_{s}$.Compute an estimate of the local probability of $B_{m}^{r_{s}}\left(i\right)$ as $C_{m}^{r_{s}}\left(i\right)=\frac{B_{m}^{r_{s}}\left(i\right)}{N-n-1},\;n=\left(m-1\right)*l$.Compute an estimate of the global probability of $B_{m}^{r_{s}}\left(i\right)$ as $\Phi_{m}^{r_{s}}=\frac{\sum_{i=1}^{N-n}C_{m}^{r_{s}}\left(i\right)}{N-n}$.Repeat step 1–5 with an increased embedding dimension, m+1, and obtain the updated global probability, denoted as $\Phi_{m+1}^{r_{s}}=\frac{\sum_{i=1}^{N-n}C_{m+1}^{r_{s}}\left(i\right)}{N-n},\;n=m*l$.Sample entropy is defined as $SE\left(m,l,r_{s},N\right)=-\ln\left[\frac{\Phi_{m+1}^{r_{s}}}{\Phi_{m}^{r_{s}}}\right]$.


**Algorithm 2.** Fuzzy entropy.Given a univariate data set ${\left\{x\left(i\right)\right\}}_{i=1}^{N}$ of length *N*, the parameters involved are the embedding dimension, *m*, tolerance, $r_{f}$, and time delay, *l*. The fuzzy function applied here is the Gaussian function with a chosen order, η.
Form the centered embedding matrix, $U_{m}$, derived from the original signal, ${\left\{x\left(i\right)\right\}}_{i=1}^{N}$, by $u_{m}\left(i\right)=x_{m}\left(i\right)-\mu_{m}\left(i\right)$, where $\mu_{m}\left(i\right)=\frac{1}{m}\sum_{k=1}^{m}x_{m}\left(i+k\right)$.Compute the distance between all the pairwise embedding vectors, $u_{m}\left(i\right)\;\;\&\;\;u_{m}\left(j\right)$, based on the Chebyshev distance, as $d_{m}\left(i,j\right)=max\{u_{m}\left(i+k\right)-u_{m}\left(j+k\right)||\;0\leq k\leq m-1,i\neq j\}$.Convert the distance matrix, $D_{m}$, to the similarity matrix, $S_{m}^{r_{f}}$, calculated through the Gaussian function as $s_{m}^{r_{f}}\left(i,j\right)=e^{\left(\frac{d_{m}{\left(i,j\right)}^{\eta }}{r_{f}}\right)}$.Compute the estimated local probability of $S_{m}^{r_{f}}\left(i\right)$ by $C_{m}^{r_{f}}\left(i\right)=\frac{\sum_{j=1,j\neq i}^{N-n-1}S_{m}^{r_{f}}\left(i,j\right)}{N-n-1},\;n=\left(m-1\right)*l$.Compute the estimated global probability of $S_{m}^{r_{f}}\left(i\right)$, by $\Phi_{m}^{r_{f}}=\frac{\sum_{i=1}^{N-n}C_{m}^{r_{f}}\left(i\right)}{N-n}$.Repeat step 1–5 with an increased embedding dimension, m+1, and obtain the updated global probability as $\Phi_{m+1}^{r_{f}}=\frac{\sum_{i=1}^{N-n}C_{m+1}^{r_{f}}\left(i\right)}{N-n}$, with n=m∗l.Fuzzy entropy is defined as $FE\left(m,l,r_{f},N\right)=-\ln\left[\frac{\Phi_{m+1}^{r_{f}}}{\Phi_{m}^{r_{f}}}\right]$.


### 3.2. Multiscale Entropy

The information provided by single-scale entropy is limited, as it does not account for long-range correlations and temporal fluctuations over multiple scales [20]. To this end, by employing the consecutive coarse-graining process (CGP) with a new parameter denoted as the scale factor, τ, Costa et al. proposed multiscale entropy [20], given in Algorithm 3.
**Algorithm 3.** Multiscale sample entropy and multiscale fuzzy entropy.Assume that a univariate data set, ${\left\{x\left(i\right)\right\}}_{i=1}^{N}$, is of length, *N*, and the coarse graining scale factor is donated as τ.
Obtain the scaled time series, ${\left\{y\left(j\right)\right\}}_{j=1}^{N/\tau }$, by the coarse graining process as
$$y^{\left(\tau \right)}\left(j\right)=\frac{1}{\tau }\sum_{i=j-\tau /2-1}^{j+\tau /2-1}x(i),\;1\leq j\leq \frac{N}{\tau }.$$Apply the scaled data set, $y^{\left(\tau \right)}\left(j\right)$, as the input into sample entropy given in Algorithm 1 or fuzzy entropy in Algorithm 2 to obtain the complexity estimation based on multiscale sample entropy (MSE) and multiscale fuzzy entropy (MFE), respectively.


### 3.3. Multivariate Multiscale Entropy

The implementation of CGP reveals the hidden information buried in multiple scales. However, the averaging process requires multiple times larger data size. To this end, multivariate multiscale entropy based on SampEn (MMSE) was proposed [19], followed by multivariate multiscale fuzzy entropy [26]. There are two main modifications in multivariate entropy algorithms based on amplitude distance. One is that the embedding vector in the univariate SampEn is replaced by the composite delay vector (CDV) composed by the embedding vectors from each channel in multivariate entropy. Figure 1 shows an example of composite delay vector construction for m=3 and l=1. In addition, due to the different range of amplitudes across channels, the obtained distance between pairwise composite delay vectors can be biased. Therefore, the second modification of multivariate entropy is the requirement of normalization of input data sets. As stated in [26], the performance of MMFE is proved to be more consistent than MMSE, particularly in high-dimensional phase spaces. The processes of MMSE and MMFE are summarized in Algorithm 4.
**Algorithm 4.** Multivariate multiscale sample entropy and multivariate multiscale fuzzy entropyGiven a multivariate data set with *P* channels ${\left\{x_{k,i}\right\}}_{i=1}^{N},\;1\leq k\leq p$, of length *N*. Parameters involved are the embedding dimension, $M=[m_{1},m_{2},\ldots ,m_{p}]$, tolerance, *r*, time delay, $L=[l_{1},l_{2},\ldots ,l_{p}]$, and the scale factor, τ.
Standardize the original multivariate data sets by subtracting the mean and dividing by standard deviation for each channel.Coarse graining is next applied to the normalized datasets, ${\left\{y_{k,i}^{\left(\tau \right)}\right\}}_{j=1}^{N/\tau }$, for each channel following Algorithm 3.Form the composite delay matrix, $Y_{M}\left(i\right)$, according to the embedding dimension, *M*, and the time delay, *L*, in the form
$$\begin{array}{ccc} Y_{M}\left(i\right) & = & [y_{1,i},y_{1,i+l_{1}},\cdots ,y_{1,i+\left(m_{1}-1\right)l_{1})},\\ & & y_{2,i},y_{2,i+l_{2}},\cdots ,y_{2,i+\left(m_{2}-1\right)l_{2})},\\ & & \vdots \\ & & y_{p,i},y_{p,i+l_{p}},\cdots ,y_{p,i+\left(m_{p}-1\right)l_{p})},] \end{array}$$Apply the scaled composite delay matrix, $Y_{M}\left(i\right)$, as the input into the sample entropy presented in Algorithm 1 or fuzzy entropy in Algorithm 2 to obtain the measure of multivariate multiscale sample entropy (MMSE) and multivariate multiscale fuzzy entropy (MMFE), respectively.


## 4. Cosine Similarity Entropy and Multi-Variate Approach

Multiscale cosine similarity entropy (MCSE) was proposed [18], whereby instead of amplitude-based distance, CSE employs the angular distance in phase space to define the difference among embedding vectors. The angular distance offers advantages, especially regarding the sensitivity to outliers or sharp changes in time series that amplitude-distance-based calculation suffers from. Additionally, the angular distance, restricted by the maximal length of 2π, is more robust, and is less prone to generating undefined estimates in the presence of noise. In amplitude-based entropy calculation, because of the large distance that is obtained by amplitude, the tolerance, *r*, is set as a ratio associated with the standard deviation of the input data set, where parameters driven by data can be unstable for signals with high variance. Given that cosine similarity entropy is restricted to values within [0,1], the tolerance for CSE is selected, irrelevant to the variance of original signal. Hence, the process of CSE exhibits enhanced stability, especially when dealing with highly dynamical signals. The virtue of CSE is therefore that entropies can be assessed more stably in multivariate analysis when the data sets across channels have different variance.

The extension of MCSE to multivariate scenarios employs the construction of the composite delay vector in MMSE [19] and follows the main procedure of CSE [18]. Note that multivariate multiscale cosine similarity entropy (MMCSE) is sensitive to the DC offset, as it requires the origin coordination to project the angular distance within the tolerance angle. Therefore, the DC offset or a long-term trend needs to be removed in MMCSE. Here, we apply the zero-median approach to remove the global trend because of its robustness against outliers. It is worth highlighting that the definition of CSE is developed based on Shannon entropy, where the calculation only relies on the probability of phase space, *m*, without the involvement of phase space m+1 [18], which is less computationally expensive. The proposed MMCSE is presented in Algorithm 5.
**Algorithm 5.** Multivariate multiscale cosine similarity entropy.Given a multivariate data set with *P* channels ${\left\{x_{k,i}\right\}}_{i=1}^{N},\;1\leq k\leq p$, of length *N*. Parameters involved are designated as the embedding dimension, $M=[m_{1},m_{2},\ldots ,m_{p}]$, tolerance, $r_{c}$, time delay, $L=[l_{1},l_{2},\ldots ,l_{p}]$, and the scale factor, τ.
Remove the DC offset of the original data sets ${\left\{x_{k,i}\right\}}_{i=1}^{N}$ by subtracting the median for each channel.Obtain the scaled time series ${\left\{y_{k,i}^{\left(\tau \right)}\right\}}_{j=1}^{N/\tau }$ following the instruction of Algorithm 3.Reconstruct composite delay vectors $Y_{M}\left(i\right)$ as demonstrated in Algorithm 4.Calculate the angular distance based on cosine similarity for all pairwise composite delay vectors, $Y_{M}\left(i\right)$ & $Y_{M}\left(j\right)$, as
$$d_{M}\left(i,j\right)=\frac{1}{\pi }\cos^{-1}\left(\frac{Y_{M}\left(i\right)\cdot Y_{M}\left(j\right)}{|Y_{M}\left(i\right)||Y_{M}\left(j\right)|}\right),\;i\neq j$$Compute the number of similar patterns defined as similar pair, $B_{M}^{r}\left(i\right)$, that satisfy the criterion $d_{M}\left(i,j\right)\leq r_{c}$.Compute the estimated local probability of $B_{M}^{r_{c}}\left(i\right)$ by $C_{M}^{r_{c}}\left(i\right)=\frac{B_{M}^{r_{c}}\left(i\right)}{N-n-1}$, where n=max(M)∗max(L).Compute the estimated global probability of $B_{M}^{r_{c}}\left(i\right)$ as $\Phi_{M}^{r_{c}}=\frac{\sum_{i=1}^{N-n}C_{M}^{r_{c}}\left(i\right)}{N-n}$.Cosine similarity entropy is defined as
$$MMCSE\left(M,L,r_{c},N\right)=-\left[\Phi_{M}^{r_{c}}\log_{2}\Phi_{M}^{r_{c}}+\left(1-\Phi_{M}^{r_{c}}\right)\log_{2}\left(1-\Phi_{M}^{r_{c}}\right)\right].$$


## 5. Value of Parameters

As pointed out in the MMCSE algorithm, several parameters need to be manually selected. This section provides a comprehensive discussion regarding the choice of the parameters, including the tolerance, *r*, embedding dimension, *m*, and data length, *N*. The corresponding performances of tri-variate MMSE and MMFE are examined against MMCSE. Throughout the analysis, the time delay was set to *L* = 1 for all simulations to avoid unknown influence and also make the temporal span fully controlled by the modification of the embedding dimension. Unless otherwise stated, the default values of parameters for MMSE and MMFE are set to *r* = 0.15∗tri(S), *M* = 2 and *N* = 10,000, where tri(S) refers to the total variance of the covariance matrix, *S*. The mean complexity curves in each subsection are plotted with error bars representing the standard deviation. Both the mean and standard deviation were calculated over outcomes of 10 independent realizations for each system.

### 5.1. Tolerance, r

Unlike MMSE and MMFE, the selection of tolerance, *r*, in MMCSE is independent of the variance of the input data sets. However, due to multiple channels, the influence of the number of variate needs to be considered when selecting the value of tolerance in MMCSE. To explore the relationship of the number of channels, *p*, and the tolerance, *r*, multi-channel white Gaussian noise (WGN) was utilized as the reference signal due to its complete randomness and simplest structure. The performances of multivariate entropy were first evaluated by varying tolerance, and then the mathematical relation was given by the best fit of the curve between *r* and *p*.

The left panel of Figure 2a shows the outcomes of multivariate single-scale CSE based on WGN as a function of tolerance for a varying number of input channels, from a univariate to 16-variate case. The tolerance was initially set to vary from 0.01 to 1 at intervals of 0.05 and was then linearly interpolated using values ranging from 0 to 1, at 0.01 intervals. Observe, in all curves, a rise in entropy with tolerance from 0 to 0.5 and a decrease in entropy with tolerance from 0.5 to 1. Due to this symmetry, only the range of 0 to 0.5 was considered to give the common relationship that the larger the tolerance, the more similar patterns are found shown as higher entropy. As discussed in [18], the tolerance for univariate multiscale cosine similarity entropy was empirically set to *r* = 0.07, whereby the performance of MCSE applied to WGN resulted in the entropy value of 0.365. Therefore, the relationship between the number of variates and tolerance, *r*, was estimated by setting the entropy as 0.365 as a standard (the black dashed line shown in the left panel in Figure 2a), as revealed in the right panel of Figure 2b using piece-wise cubic interpolation fitting curve. It is found that the best fit was for
(1)$$r_{cse}=-0.4\left(p^{-0.71}\right)+0.47,$$
where *p* donates the number of variates.

Therefore, the tolerance in MMCSE is decided as a function of the number of variates. This relationship is given in Equation (Equation 1) to maintain the performance of multivariate CSE for a varying channel number. For the commonly applied situation when the channel number is 2 (bivariate data), the typical setting value of tolerance is *r* = 0.225 and for the tri-variate case, it is *r* = 0.287, according to the equation.

### 5.2. Embedding Dimension, m, and Data Length, N

The embedding dimension, *m*, in amplitude-distance-based methods, is usually restricted by the data length, *N*. As a rule of thumb, *N* should scale as $10^{m}$ [29,30]. For fuzzy entropy, the limitation is relaxed such that *N* should exceed 50 samples to give a defined estimate. That is due to the continuous boundary of the transfer function between similarity and tolerance at the expense of computational load [16]. As a result, limited by the sample size and computational time, the structure of complexity in high embedding dimensions is usually poorly estimated by current entropy methods. The trade-off between the embedding dimension and sample length means that the entropy methods are difficult to implement in real time. The influences of the embedding dimension, *m*, and the data length, *N*, of these multivariate entropy methods are discussed here.

The performance of the three multivariate entropy algorithms, MMSE, MMFE, and MMCSE, was tested on white Gaussian noise (WGN) and autoregressive (AR) models AR(1), AR(2), and AR(3) in three-channel multivariate systems. A total of 10 independent realizations were generated for each model with a constant sample size of *N* = 2000 points for all operations, and the results are shown in Figure 3. The mean entropies are shown, and error bars designated the standard deviation. Coefficients of the considered AR models are given in Table 1.

The mean values of entropy are shown as a function of the embedding dimension, *m*, in Figure 3. The given embedding dimension ranges from 1 to 10 with increments of 1 for MMSE and MMFE, and the range of used embedding dimension for MMCSE was set to 2 to 10, due to the minimum requirement of 2 when calculating angular distance. Observe that in the left panel of Figure 3, the selection of *m* for MMSE is limited to lower than 7 to give a defined estimation for the most complex signal AR(3). On the other hand, for uncorrelated WGN, the maximum embedding dimension with a defined positive MMSE value is *m* = 2. In the second panel, MMFE produces a more stable estimation at higher scales, in contrast to MMSE. Yet, the differences between the four synthetic signals decrease with increasing the embedding dimension, and when *m* comes to 9, the complexity estimates overlap. In addition, resulting from the single-scale analysis, both MMSE and MMFE fail to order the complexities of the four simulated signals. The highest complexity among the given models ought to be assigned to AR(3). In comparison, the MMCSE in the last panel gives a positive, defined estimate for all the scenarios with the correct rank based on the structural complexity of the signals, where complexity(AR(3)) > complexity(AR(2)) > complexity(AR(1)) > complexity(WGN). Moreover, MMCSE illustrates a consistent estimation with a good separation among the four models. Therefore, MMCSE exhibits a more stable and consistent performance when applying complexity evaluation under high embedding dimensions in these simulations.

The second significant parameter that controls the performance of the complexity estimation based on entropy is the measured data length, *N*, which is difficult to control for many real-world signals. Figure 4 exhibits the performance of the three single-scale multivariate entropy methods. The default parameters were set to *M* = [2,2,2], *L* = [1,1,1], and the tolerance values were selected following the Equation (Equation 1). In the left panel in Figure 4, multivariate sample entropy performs poorly when *N* < 60 on the most complex signal here, AR(3). Generally it gives wide error bars with little separation between the four models. Multivariate fuzzy entropy and multivariate cosine similarity entropy exhibit a more stable estimate and a less strict requirement for the minimal data length compared to that given by multivariate sample entropy. However, in the middle panel and left panel, the most structurally complex model, AR(3), is wrongly assigned the lowest degree of complexity. The improvement of fuzzy entropy from sample entropy comes at the expense of calculation efficiency with more time consumption required. The cosine similarity entropy-based algorithm manages to reduce the minimal data length required without increasing the computational load.

### 5.3. Complexity Profile of MMCSE, MMSE and MMFE

The complexity profiles of multivariate multiscale entropy algorithms were first calculated for simulated linear and nonlinear signals. The same auto-regressive (AR) models were applied together with a commonly encountered nonlinear signal, the 1/f noise. With the default parameters set to *M* = [2,2,2] and *N* = 10,000, the mean complexity of 10 independent realizations with three channels are illustrated in Figure 5. Error bars represent the standard deviation over the 10 realizations.

From the left to the right panel in Figure 5, the complexity profiles of the multivariate multiscale sample entropy (MMSE), multivariate multiscale fuzzy entropy (MMFE) and multivariate multiscale cosine similarity entropy (MMCSE) are presented. With the same setting, MMSE is observed to be capable of distinguishing signals with self-correlations, including AR models and 1/f noise, from the completely uncorrelated WGN (in blue line). However, the random WGN is wrongly assigned to exhibit a structural complexity at low scales, and the separation among correlated signal is also difficult to be achieved as overlapped error bars shown across all the scales by MMSE. In contrast with MMSE, MMFE correctly demonstrates the complexity difference of the five models (AR models from WGN and 1/f noise from WGN respectively) with a narrower variance.

Observe from the middle panel in Figure 5, that the 1/f noise is expected to be a truly complex signal across all scales, which is correctly given by MMFE. However, the problem of the wrong estimation of the complexity of uncorrelated WGN still remains by applying MMFE at small scales. When estimated by MMCSE in the right panel in Figure 5, the complexity measures based on WGN remain the lowest across all scales, reflecting the uncorrelated structure of an ideal random signal. In regard to the AR processes with different orders, a system with the higher order means more coefficients involved when generating signals associated with a higher degree of freedom. Hence, the correct relation of structural complexity of simulated AR processes is supposed to be AR(3) > AR(2) > AR(1), which can be only observed in the graph given by MMCSE. Meanwhile, the estimation of the complexity given by MMCSE is consistent in the comparison between AR processes with various orders, and between 1/f noise and WGN based on self-correlation. To this end, the proposed multivariate multiscale cosine similarity entropy yields stable estimates at a high coarse-graining scale, thus making it possible to examine structural complexity of real world processes with long-range correlations.

## 6. Detection of Circularity

We next examine the application of multivariate entropy in the detection of complex and quaternion non-circularity, that is, the rotation dependence of probability density function (pdf). The non-circularity manifests itself through a degree of correlation and power imbalance of the system channels, which can be reflected in a scatter plot as the inclined angle and dispersed degree of the distribution. The long-range correlation of a time series measured by multiscale entropy reflects the influence of past states on the generation of future signals [31], while the correlation of a multi-channel system is a measure of association among channels, a summary of dependence strength for multivariate systems, and a feature that could reflect the structural richness of the system [32,33]. A strongly correlated system is expected to show a high structural complexity, while a less correlated system would approach the performance of WGN to give a low structural complexity. As for the other critical property, the power of the signal has been widely used as an indicator of dynamics in complex systems, such as the analysis based on heart rate variability in physical systems [34], where the ratio of the low-band frequency power and high-band frequency power was used to reveal the sympatho-vagal balance of the autonomous nervous system [35].

The ability of standard MMSE, MMFE and the proposed MMCSE in the detection of circularity of multivariate signals is next simulated and discussed. We consider the case of a tri-variate (pure quaternion) signal, while the analysis based on bi-variate (complex valued) system can be found in Appendix A. Systems constructed by tri-variate correlated WGN, Y, are discussed here and generated by the correlation matrix, C, shown in Equations (Equation 2) and (Equation 3). The correlation behavior is jointly controlled by the coefficients, *p* and *q*, while the correlated power is determined by the factor, *q*, that is
(2)Y=X·chol(C)
(3)$$C=\left[\begin{array}{ccc} 1 & pq & pq\\ pq & q^{2} & pq^{2}\\ pq & pq^{2} & q^{2} \end{array}\right],\;chol\left(C\right)=\left[\begin{array}{ccc} 1 & pq & pq\\ 0 & q\sqrt{1-p^{2}} & \frac{pq\sqrt{1-p}}{\sqrt{1+p}}\\ 0 & 0 & q\sqrt{1-\frac{2p^{2}}{1+p}} \end{array}\right]$$

Here, X denotes a tri-variate uncorrelated WGN system, $X=[\varepsilon_{1},\varepsilon_{2},\varepsilon_{3}]$, ε(i)∼N(0,1), and Chol(C) refers to the Cholesky decomposition based on the correlation matrix, C.

The tri-variate systems with varying correlation degree, *p*, and power, *q*, were analyzed by MMSE, MMFE, and the proposed MMCSE. The default parameters were set to *M* = [2,2,2], *L* = [1,1,1], *N* = 1000, and the maximum scale factor was chosen as τ = 10. The tolerance, *r*, for MMSE and MMFE was set as 0.15∗tr(S), where tr(S) refers to the total variance of the input signals. The choice of tolerance for MMCSE was calculated according to Equation (Equation 1). The error bars designate the average and standard deviation over 30 realizations for each model.

### 6.1. Correlated WGN with Equal Power

Tri-variate input signals associated with five different correlation degrees and equal power were first generated as scatter plots shown in Figure 6. The first model in the left graph with *p* = 0, *q* = 1, gave the uncorrelated WGN system. The four considered cases were set as the fixed power, *q* = 1, and the correlated relations among input channels, Y, were hence determined by one varying coefficient, p∈{0.4,0.6,0.75,0.99}. Observe the varying degree of non-circularity, from the rotation-invariant (circular) case on the left panel through to a high degree of non-circularity (narrow scatter plot) on the right-most panel.

Figure 7 illustrates the performance of the multivariate multiscale analysis based on the sample entropy, fuzzy entropy and cosine similarity entropy on the task of the detection of the degree of non-circularity in tri-variate (quaternion) data. Observe that MMSE and MMCSE, in the left and right panels respectively, were capable of distinguishing between the varying settings of coefficients, *p*, shown as different trends of lines, while MMFE in the middle panel was only prone to separating the case with maximum correlation (in amber) from the rest of the scenarios. As *p* increases, the structural richness of the system (degree of non-circularity) rises, as shown in Figure 6. Hence, the structural complexity for signals with large *p* is supposed to be greater than that of signals generated from small *p*, which is correctly evaluated by both MMSE and MMCSE. However, as scale, τ, increases, MMSE exhibits a descending trend and the gaps among curves narrow down accordingly, whereby the ability of MMSE to classify varying correlations is negatively influenced by the increasing scale. On the other hand, for MMCSE in the right panel, the increasing scale has no impact on the separation of different models as shown by the flat lines.

### 6.2. Uncorrelated WGN with Unequal Power

Next, five models of tri-variate (pure quaternion) uncorrelated WGN with unequal power were simulated and analyzed by multivariate entropy. Figure 8 demonstrates the scatter diagrams of five uncorrelated input systems, where the dynamics in one channel is independent of the change of others. The coefficient, *p*, was fixed as zero (to make the chol(C) a diagonal matrix) and *q* was varying in this case among {1,0.6,0.45,0.35,0.1} to control the power of the system. As shown, the above two channels exhibit no correlation, with flat scatter diagrams and the power imbalance increase from the left to the right when the power reaches its maximum and is fully balanced, which is the case of white noise as given in the first graph. Figure 9 shows the performance of the three multivariate entropy methods.

The left and middle panels of Figure 9 give the estimation by MMSE and MMFE, which fail to reflect the effects of different powers of the signals due to data normalization, as given in the first step of amplitude distance-based algorithms shown in Algorithm 4. In other words, the information about power imbalance is lost when normalizing the data for SampEn and FuzzyEn-based analyses, thus resulting in a decrease in entropy values approaching the behaviors of random white noise. However, data normalization is one of the key pre-processing steps for amplitude-based entropy analyses, which cannot be avoided. In contrast, the MMCSE in the right panel is observed to detect the discrepancy of powers among the data channels of the uncorrelated systems, resulting from non-unified variance towards the input multivariate data sets based on the angular distance. In addition, the non-descending trends in the right panel remain constant, irrespective of the varying power, where the expected performance of structural complexity estimation can be observed at single scale by multivariate CSE.

### 6.3. Correlated WGN with unequal power

Next, we jointly considered the two above-discussed scenarios for systems of tri-variate (pure quaternion) correlated WGN associated with unequal powers in data channels. In terms of the correlated input, the scatter diagrams are illustrated in Figure 10, where *q* is varying within {1,0.5,0.35,0.1}, given *p* a non-zero value as *p* = 0.6. For comparison, the uncorrelated WGN, *p* = 0 & *q* = 1, was also produced as the model with least structural complexity (circular) as shown in the first diagram in Figure 10. Figure 11 demonstrates the complexity estimation based on the models generated.

With a fixed, non-zero coefficient *p*, the increasing power imbalance (non-circularity) yields a higher structural complexity with stronger correlation as shown in the scatter plots from the left to the right in Figure 10. Regarding the complexity measures, observe that MMSE in the left panel of Figure 11 could only separate the completely uncorrelated signals (in blue) from the correlated signals, while MMFE fails to detect the significant discrepancy of structural complexity within all the simulated models. In terms of the proposed MMCSE, signals with various powers and correlation structures (different degrees of non-circularity) are separated in a consistent way in the right panel in Figure 11. To further explore the ability of complexity estimation for signals with jointly varying coefficients *p* and *q*, the heatmaps of entropy estimated via multivariate single-scale sample entropy, multivariate single-scale fuzzy entropy and multivariate single-scale cosine similarity entropy are displayed in Figure 12, where *p* in the *y*-axis was set to range from 0.05 to 0.99 at 0.05 intervals, and power, *q*, in the *x*-axis ranged from 0.05 to 1 at 0.05 intervals. Observe that SampEn- and FuzzyEn-based multivariate analyses are sensitive to the correlation coefficient *p*, but are insensitive to signal power, *q*. Comparing the two amplitude entropy methods, multivariate SE exhibits higher sensitivity to changes of *p* than multivariate FE. In contrast, the proposed CSE-based multivariate method manages to detect changes in both *p* and *q* as shown in the right heatmap in Figure 12.

## 7. Conclusions

This work extended the univariate cosine similarity entropy to the multivariate case. It was shown that the proposed multivariate multiscale cosine similarity entropy (MMCSE) method is capable of estimating structural complexity based on self-correlation with higher stability at large scales and lower requirement on data length than the existing methods. In contrast with standard MMSE and MMFE, the MMCSE has also exhibited higher consistency in multiscale analysis, where the long-range correlation of signals can be correctly measured at low scales. The performance of MMCSE was first examined on five benchmark signals to reveal the improved estimation against MMSE and MMFE, and then tested on multivariate correlated systems associated with unequal power among data channels (complex and quaternion non-circularity), to detect the properties of circularity including the degree of correlation among channels and the strength of power in the system. As desired, due to the angle-based distance and the relaxation of the requirement of data normalization, the proposed MMCSE exhibited noticeable improvement in detecting the correlation degree of the multivariate system and also showed the unique feature to detect the change of power in the system, unlike MMSE and MMFE. The results indicated the wide range of potential applications of MMCSE in complexity science for practical scenarios. However, regarding the limitation of the coarse-graining average, the multiscale procedure can be further improved by employing the enhanced scaling process. It should be noted that this study is restricted to the analysis on synthetic signals. Therefore, future work will focus on the analysis of real-world data sets and involve statistical methodologies.

## Figures and Tables

**Figure 1 entropy-24-01287-f001:**
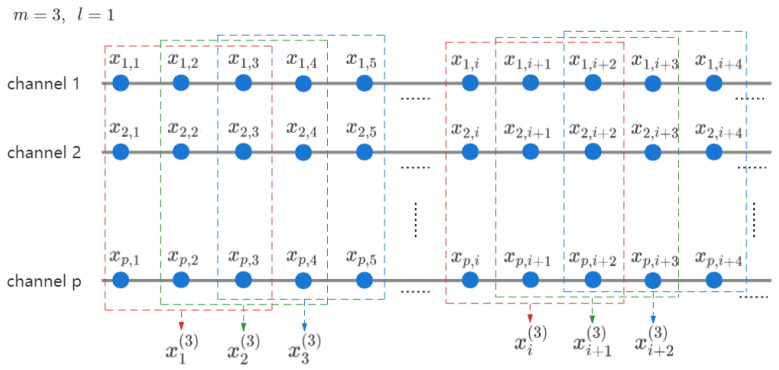
Example of composite delay vectors construction in multiscale entropy.

**Figure 2 entropy-24-01287-f002:**
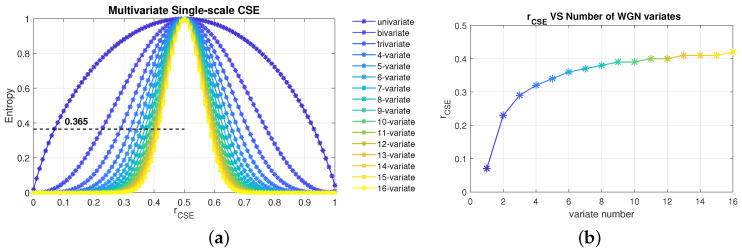
Choice of parameters in MCSE. (**a**) Behaviour of multivariate single-scale CSE on the estimation of white Gaussian noise (WGN), as a function of the tolerance, *r*. The default parameters were set as *N* = 10,000, *M* = 2, and *L* = 1 for all channels. The error bars designate the standard deviation over 10 realizations. (**b**) The optimal tolerance, *r*, as a function of the number of variates.

**Figure 3 entropy-24-01287-f003:**
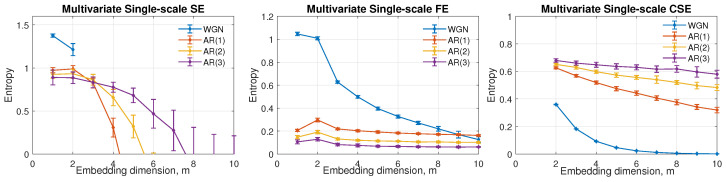
Behavior of multivariate single-scale SE, FE and CSE as a function of the embedding dimension, *m*. The default parameters were set as *N* = 2000, *L* = [1,1,1], *r* = 0.45 for multivariate SE/FE, and *r* = 0.287 for multivariate CSE.

**Figure 4 entropy-24-01287-f004:**
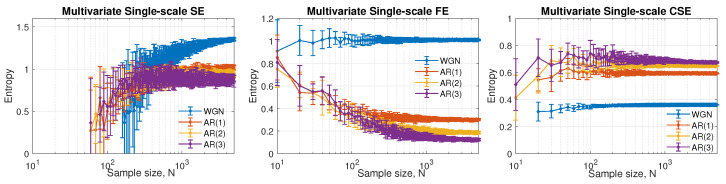
Behavior of multivariate single-scale SE, FE and CSE as a function of the data length, *N*. The default parameters were set as *M* = [2,2,2], *L* = [1,1,1], *r* = 0.45 for multivariate SE/multivariate FE, and *r* = 0.287 for multivariate CSE.

**Figure 5 entropy-24-01287-f005:**
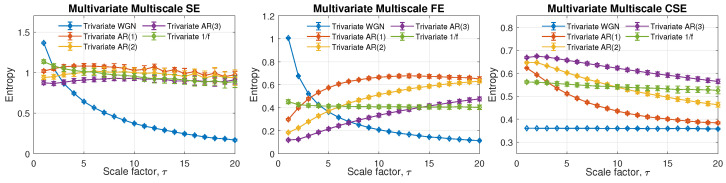
Complexity profile of MMSE, MMFE, and MMCSE. The default parameters were set as *M* = [2,2,2], *N* = 10,000, *L* = [1,1,1], *r* = 0.45 for MMSE/MMFE, and *r* = 0.287 for MMCSE.

**Figure 6 entropy-24-01287-f006:**
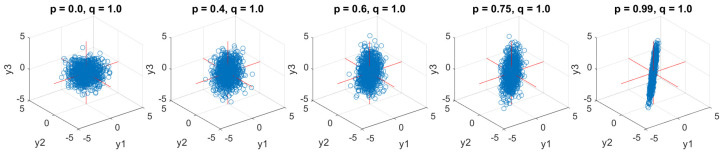
Scatter diagrams of tri-variate (quaternion) input correlated WGN (pure quaternion) with equal powers in the three data channels, but different degrees of correlation among them. Observe an increase in the degree of non-circularity from left to right.

**Figure 7 entropy-24-01287-f007:**
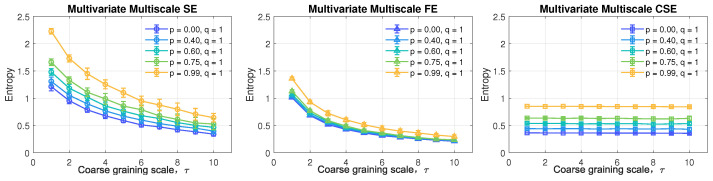
Behavior of standard MMSE, MMFE, and the proposed MMCSE applied to correlated tri-variate (quaternion) WGN with equal power.

**Figure 8 entropy-24-01287-f008:**
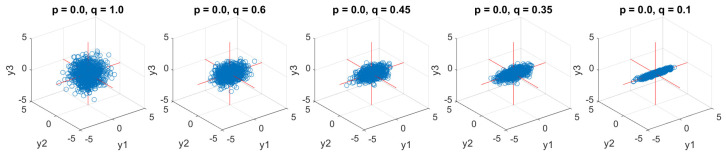
Scatter diagrams of tri-variate (pure quaternion) input uncorrelated WGN with unequal powers in data channels. Observe the increase in the degree of non-circularity, from the left to the right panel, with the increase in power imbalance between the data channels.

**Figure 9 entropy-24-01287-f009:**
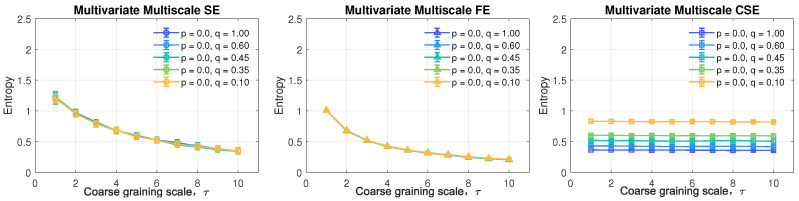
Behavior of standard MMSE, MMFE, and the proposed MMCSE applied to tri-variate (quaternion) uncorrelated WGN with unequal powers in data channels.

**Figure 10 entropy-24-01287-f010:**
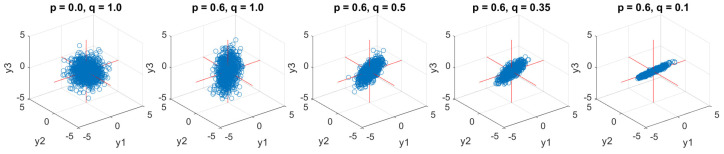
Scatter diagrams of tri-variate (quaternion) input correlated WGN with unequal power.

**Figure 11 entropy-24-01287-f011:**
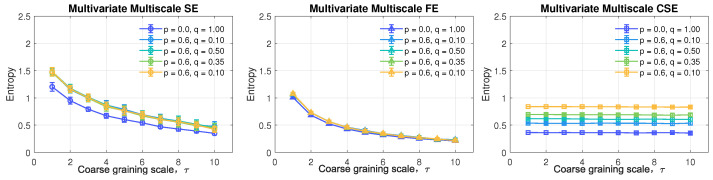
Behavior of standard MMSE, MMFE, and the proposed MMCSE applied to correlated tri-variate (quaternion) WGN with unequal power.

**Figure 12 entropy-24-01287-f012:**
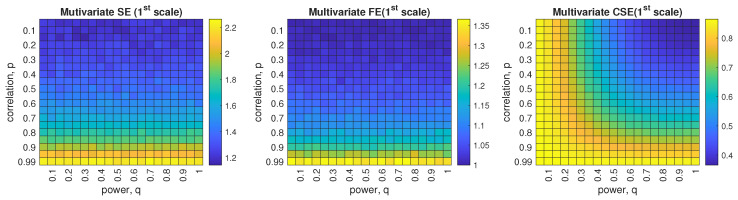
Heatmap of complexity estimation based on standard multivariate SE, multivariate FE, and the proposed multivariate CSE in a function of varying coefficients *p* and *q*.

**Table 1 entropy-24-01287-t001:** Coefficients of the AR models used.

Coefficients	$a_{1}$	$a_{2}$	$a_{3}$
AR (1)	0.9	−	−
AR (2)	0.7	0.25	−
AR (3)	0.6	0.25	0.125

## Data Availability

The data that support the findings of this study are openly available in figshare at https://doi.org/10.6084/m9.figshare.21084355.v1 (accessed on 11 August 2022).

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
