# Peer review of "Multivariate Multiscale Cosine Similarity Entropy and Its Application to Examine Circularity Properties in Division Algebras†"

_entropy, 2022, doi:10.3390/e24091287_

Round 1

Reviewer 1 Report

Dear Authors, your work Is very good.

May I suggest the following reference to be considered:

Morabito et al, Multivariate multiscale permutation entropy for complexity analysis in AD, Entropy, 2012

This paper is highly innovative, the methodology proposed is novel, the results achieved are highly encouraging.

Author Response

Dear Reviewer:

Many thanks for your positive and insightful comments. We have augmented the literature review accordingly with the recommended reference paper, which can be found in Reference [2].

Reviewer 2 Report

A number of points are required to improve the article for the publication process:

The introduction is too long, try to separate the “Related Work” into a new section.

Highlight the problem statement and main contribution inside the introduction.

In algorithm 4, how does the data set length affect the number of channels P? please clarify it.

Discuss the entropy values that exceed the “1” value in figure 3.

Figures 8, and 9 need more explanations.

Reviewer 3 Report

The authors extended cosine similarity entropy to multivariate case to estimate structural complexity with self-correlation, compared their proposed method with standard methods like MMSE and MMFE. It analyzes behavior from several different aspects including dimensions, AR processes, noise tolerance, etc. Overall, the paper is written with very clear presentation. 

Author Response

Dear Reviewer:

Many thanks for your positive comments.